# Firewall Best Practices for Securing Smart Healthcare Environment: A Review

**Raja Waseem Anwar** [1,*] **, Tariq Abdullah** [2] **and Flavio Pastore** [2]

[1] Faculty of Computer Studies, Arab Open University, P.O. Box 1596, Muscat 130, Sultanate of Oman
[2] College of Science & Engineering, University of Derby, Derby DE22 1GB, UK; t.abdullah@derby.ac.uk (T.A.); f.pastore1@unimail.derby.ac.uk (F.P.)
[*] Correspondence: waseem@aou.edu.om; Tel.: +968-92336043

**Abstract:** Smart healthcare environments are growing at a rapid pace due to the services and benefits offered to healthcare practitioners and to patients. At the same time, smart healthcare environments are becoming increasingly complex environments where a plethora of devices are linked with each other, to deliver services to patients, and they require special security measures to protect the privacy and integrity of user data. Moreover, these environments are exposed to various kinds of security risks, threats, and attacks. Firewalls are considered as the first line of defense for securing smart healthcare networks and addressing the challenges mentioned above. Firewalls are applied at different levels in networks, and range from conventional server-based to cloud-based firewalls. However, the selection and implementation of a proper firewall to get the maximum benefit is a challenging task. Therefore, understanding firewall types, the services offered, and analyzing underlying vulnerabilities are important design considerations that need addressing before implementing a firewall in a smart healthcare environment. The paper provides a comprehensive review and best practices of firewall types, with offered benefits and drawbacks, which may help to define a comprehensive set of policies for smart healthcare devices and environments.

**Keywords:** firewall; smart healthcare; vulnerabilities; cloud; security

## 1. Introduction

The integration of Internet of Things (IoT) with smart healthcare devices brought many opportunities and challenges. Healthcare professionals and medical practitioners can monitor their patients remotely and deliver critical medical care using these devices. The digitization and rapid increase of smart healthcare environments offer a wide range of benefits that not only help healthcare professionals and healthcare providers, but also offer better, and around the clock service to patients. Moreover, patients can be monitored remotely, and healthcare professionals can share their experiences in real-time through these interconnected environments. In addition to traditional "information systems" implemented in healthcare networks, the growth of the IoT paradigm allows various mobile devices and sensors to be part of a healthcare network for monitoring these systems and patients in real-time, but also make these networks more vulnerable and increase the probability of cyberattacks [1]. Such attacks usually target sensitive healthcare records that not only affect the safety and privacy of patients, but also threaten the integrity of the data. Each day, thousands of healthcare systems are threatened worldwide, and "invisible" attackers cause irreversible damages. For example, in 2017, the average cost of a single cyberattack in the UK was $8.7 million, while it was $11.5 million in 2018 [2], registering an increase of 31%. Therefore, it is equally important to have proper management and continued monitoring for all healthcare entities. Cybersecurity is an economic, military, and social issue, and securing healthcare systems can essentially be resumed by the respect of three properties: integrity, confidentiality, and availability [3,4].

Firewalls play a critical role in securing and accessing electronic health records and network-enabled medical devices across and beyond the network where these services are accessible through the cloud [5]. In fact, firewalls have always been the first line of defense for healthcare system security. To ensure those properties, a good firewall must always act as a filter between the traffic coming from internal networks (outgoing traffic) and the traffic generated from outside and generally less secure networks (ingoing traffic) than protecting networks from external attacks and risks, with consequent enforcement of security, and no restricted policy to allowed users. The effectiveness of a firewall system depends on several environmental conditions and constraints, such as the placement of a firewall and the nature of data that need to be protected. Moreover, users should always be able to access resources behind the firewall without facing any issues [6,7]. Smart healthcare architectures are like any other network systems, in general, yet considering the data contained in these systems, the architecture is largely different from other similar types of architecture. More specifically, considering the life-critical data of a patient, financial implications and data governance compliance requirements require additional layers of security for smart healthcare architectures, and this makes these architectures different from other similar network systems.

The landscape of cybersecurity threats and new security vulnerabilities to smart healthcare systems is constantly growing. The effective and efficient use of firewalls reduces the impact of cyber threats on smart healthcare environments. More specially, cloud-based firewalls play an important role while safeguarding cyber-attacks against smart healthcare devices carrying sensitive data and information. Therefore, it is important to have proper firewall placement for a smart healthcare environment to protect against attacks and threats.

Despite providing the various advantages and benefits in securing healthcare systems, firewalls are still vulnerable to attacks. In this paper, we investigate the most known firewall types and their compatibility to smart healthcare environments. Secondly, we provide a brief description of various types of firewalls, along with their working mechanisms and vulnerabilities for these environments. Thirdly, we identify and evaluate the firewall best practices and placement strategies that help in selecting a firewall suitable for smart healthcare environments. Lastly, we summarize the open challenges and provide recommendations for securing smart healthcare environments. The selection of a proper firewall for smart healthcare will not only protect the confidentiality and privacy of patient data, but also improve trust and security.

The rest of the paper is organized as follows. Section 2 provides a review of security challenges for a smart healthcare environment. Section 3 describes the firewall characteristics related to smart healthcare environments. Section 4 describes the firewall placement for these environments. Section 5 highlights the firewall vulnerabilities followed by the firewall best practices for a smart healthcare environment in Section 6. Similarly, discussion and open challenges are presented in Section 7. Conclusions from this work are provided in Section 8.

## 2. Security Challenges for Smart Healthcare Environment

In this section, we provide the security challenges for smart healthcare environments. Before describing the security challenges, we would like to provide a typical smart healthcare architecture.

During the past decade, there have been tremendous advancements and developments in the field of information and communication technologies (ICTs), providing many benefits to society. Moreover, these intelligent and interconnected systems, where information is constantly shared between hospitals, doctors, and patients, require proper monitoring and safeguarding against threats and attacks [8]. Healthcare has always been a very important part of our lives, as the patients' health can be ensured by using the services of medical practitioners and the healthcare industry. Smart health monitoring devices provide facilities for paramedics to monitor and track patients remotely and provide relevant

diagnostics [9]. Smart healthcare systems, which consist of various components, such as smart sensors, Radio Frequency Identification (RFID) devices, are constantly sending medical data from patients to local workstations located at the hospital [10]. However, when dealing with patient data, the underlying privacy on the data requires proper safeguarding before transmitting via interconnected network components. A typical smart healthcare architecture is presented in Figure 1. The integration of smart healthcare devices between hospitals, paramedics, and the internet is facing significant cybersecurity challenges. Smart healthcare devices are connected to share data and are linked with other devices using the internet, to upload and download data to and from cloud servers. Therefore, the placement of a specific firewall is the key component that defends against internal and external attacks and threats.

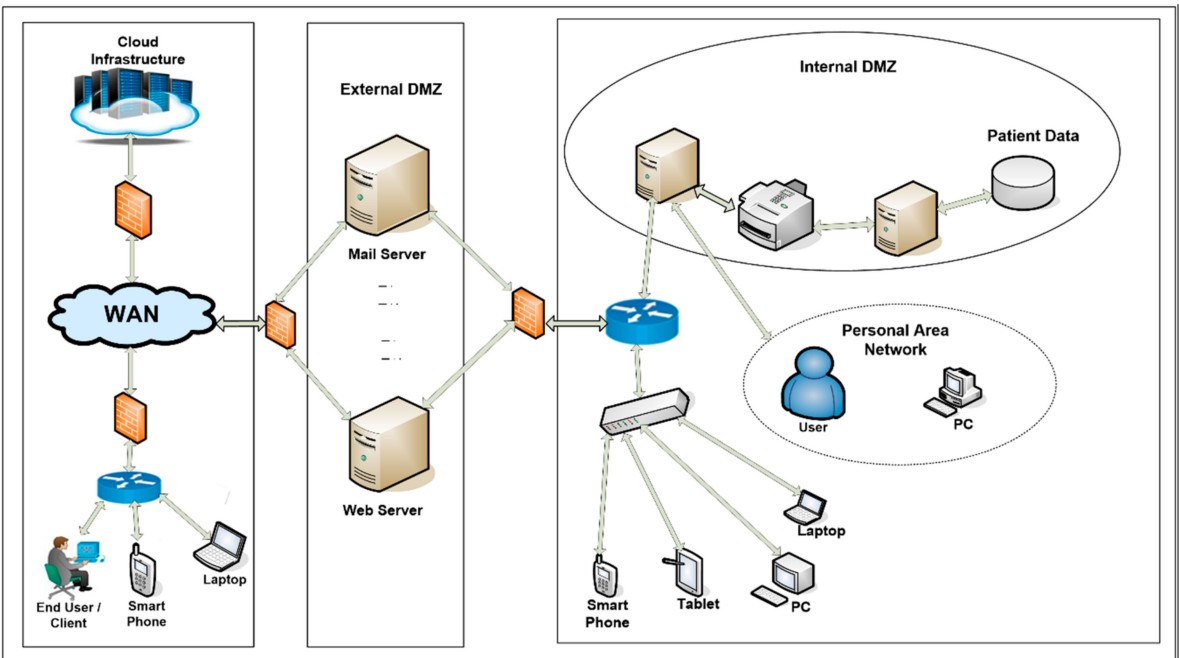

**Figure 1.** Smart Healthcare Architecture.

The demand for healthcare systems is growing because of aging populations and an increase in chronic diseases [11]. Moreover, the adoption of online healthcare advice and the services provided are heavily dependent on connected devices and network services. Healthcare systems become a key target for cybercriminals due to the value of the information they contain; therefore, security is a must for healthcare devices to protect sensitive data and critical infrastructure [12,13]. Lack and poorly implemented security in healthcare applications have negative impacts and consequences on patient data, privacy, and vulnerability to various attacks. Therefore, failure in maintaining the security for healthcare devices can result in the loss of integrity, availability, and data, which ultimately risks patients' lives [14,15].

*Attacks on the Smart Healthcare Environment*

The ubiquitous computing nature of healthcare related applications and entities relying heavily on continued monitoring of patients (either living at home or treated in a hospital), unsecure systems and applications pose severe security vulnerabilities. Therefore, security is one of the most essential requirements in the healthcare environment and the underlying applications, In addition, connected devices share data remotely with doctors and other paramedics, which require proper security and privacy of communicated data and information. Moreover, in remote healthcare monitoring environments, rapid decision-making is essential specifically during emergency situations [16]. In addition, the

integration of IoT with smart healthcare devices and the environment provides efficient interaction of healthcare devices and medical databases [17]. Use of smart healthcare provides numerous advantages, but at the same time, is under the threat of various attacks, such as denial of service (DoS), eavesdropping, and message tampering attacks. The following sub-sections provide brief details of these attacks.

Cyber-attacks are a rising concern for smart healthcare environments and are considered as the most vital and demanding issues [18]. It is very critical for smart healthcare devices to deliver the services uninterrupted, as the unavailability of any service could have negative consequences on the life of the patient. Some common attacks to the smart healthcare environment are summarized as [19–21]:

- Denial of Service (DoS): in this attack, the smart healthcare environment and network communication channels are flooded with bogus messages, which ultimately consume the resources.
- SQL Injection attack: attackers inject the SQL queries into the database and extract the required information, or even delete and corrupt the database.
- Eavesdropping: under this attack, the adversary monitors the healthcare systems and obtains sensitive information.
- Malwares: in this attack, the adversary exploits known vulnerabilities in both the hardware and software. The most common types of malware are viruses, worms, and Trojan horses, in which a malicious piece of software causes these attacks, and where the attacker or adversary sends malicious code, which executes on the victim's computer, and puts the victim's computer at risk.
- Reconnaissance attack: in this attack, the adversary will gather all of the required information regarding the target.

## 3. Firewalls Types for Smart Healthcare Environment

Firewalls today represent the first line of defense against major attacks, affecting both traditional and modern networks, and enforcing the protection of inside networks from external (and untrusted) networks. The application of an effective set of security practices and policies may indeed keep those systems safe and save entire businesses. Firewalls have a very important function of protecting, filtering, and controlling all traffic sent and received from the computer, Local Area Network (LAN), or Wide Local Area Network (WLAN) internal networks from unauthorized intrusions or external attacks [22]. Firewalls can either be hardware or device software, and filter and control information coming from the internet to one's private network or individual computer. They can also filter and control information coming out from the smart healthcare networks to the internet. Firewalls allow the transit in and/or out only to the data considered safe and harmless. Data are allowed through a series of internet protocols (TCP/IP, UDP, ICMP, etc.), in which each computer is identified with an IP address. These data are also called "data packets", and these packets have the following headers that identify them:

- IP address of the packet sender;
- IP address of the packet receiver;
- Packet type (TCP/IP, UDP, etc.);
- Port number (port is a number related to a service/network application). The standard allowed ports are typically 21/FTP, 22/SSH, 23/TELNET, 80/HTTP, and 443/HTTPS.

Usually, a received packet is matched against a certain rule previously set; this can be discarded or forwarded depending on what this rule establishes. This function introduces the first and basic firewall category, packet filtering. This method allows dynamically generating packet filter rules, showing a certain expiration time instead of being static (or stateless), which is the stateful inspection firewalls. Regarding the smart healthcare environment, the selection of a firewall with an appropriate configuration of rules effectively reduces the impact of attack [23]. We can have hardware firewalls, which are physical devices used to protect local networks, and are normally integrated into TCP/IP routers or software firewalls, applicable to public networks using a central computer installing

firewall software. Firewalls act as a filter on IP address, packet attributes, or a connection state basis, according to the Open Systems Interconnection (OSI) layer classification. Figure 2 presents the broader classification of various protocols and firewall types, which are arranged according to the OSI reference model [24].

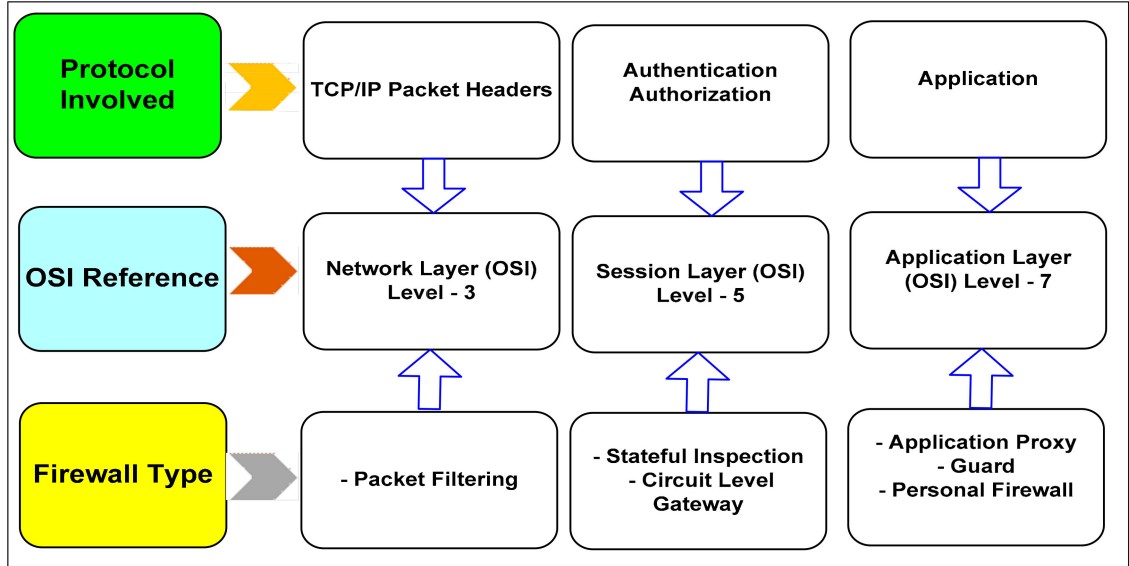

**Figure 2.** Firewall practice and level of controls according to the OSI reference model.

Moreover, the firewalls are often grouped into network firewalls and host-based firewalls [25]: the former generally run on dedicated hardware and can handle data from (and to) a large number of servers on one or multiple networks. The latter are configured directly on a host or cloud server, limiting themselves to controlling the traffic generated and delivered to the machine. Granting access to multiple developments, pre-production, and production environments, in which, through automatic systems and pipelines, code, and configurations through fast and streamlined processes, requires smarter and effective firewall rules. This first classification helps to introduce some useful characteristics to distinguish traditional firewall types from cloud firewall ones. The firewall types and their specific categories are given in Figure 3.

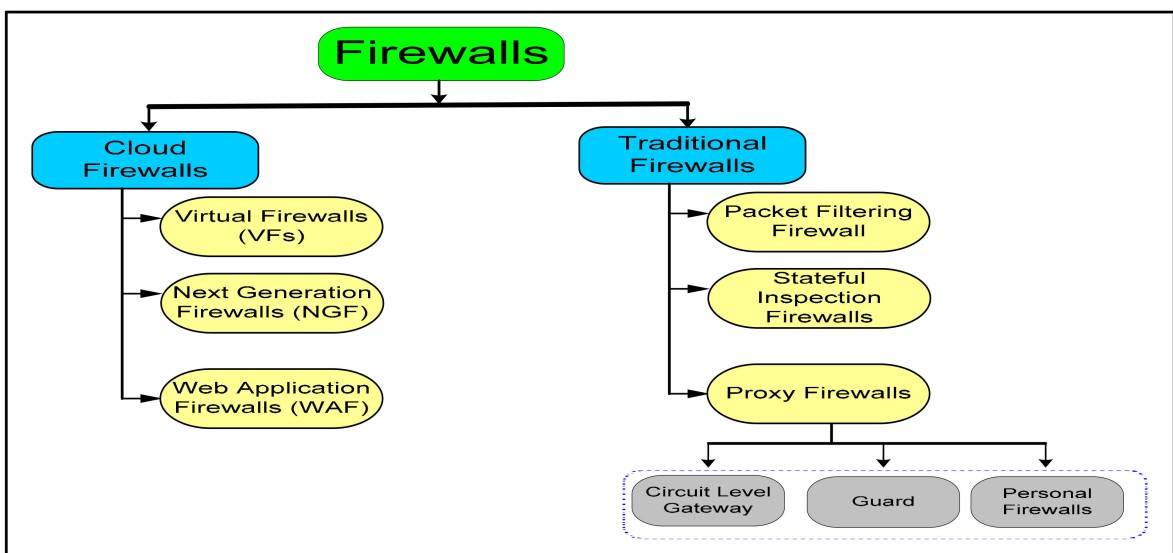

**Figure 3.** Taxonomy diagram of firewalls (micro categories).

There is growing demand for smart healthcare leveraging cloud-based computing and cloud services that offer an array of benefits in compression to in-house network systems. Moreover, the use of cloud-based firewalls reduces operational cost and influences more economic benefits. Furthermore, cloud-based firewalls for healthcare systems provide better security and privacy for health data and systems [26]. Cloud-based firewalls enhance smart healthcare systems due to the interoperability and integration and ease of connecting the systems. However, the growing demand for cloud networks require implementation of at least three additional firewall types:

- Virtual firewalls;
- Next-generation firewalls;
- Web application firewalls.

The National Institute of Standards and Technology (NIST) Special Publication 800-10 categorizes firewalls into three types [27]:

- Packet filtering firewalls;
- Stateful inspection firewalls;
- Proxy firewalls (three sub-categories).

### 3.1. Cloud Firewall Types

The integration of smart healthcare environments with cloud-based servers provides a variety of services and benefits and allows patients to access various resources. Similarly, cloud-based firewalls are the new generation of firewalls intended for cloud networks and are able to filter traffic between virtual machines and network devices more specifically to smart healthcare devices [28]. For example, Fire wall-as-a-Service (FWaaS), typically applied to software as a service (SaaS) delivery models, is straightforward to configure, does not need physical or frequent manual updates (because it is integrated into the same platform as an all-in-one software), and no advanced knowledge of traffic is required for cloud users (who are not responsible for firewall and security policies). This is the typical case of FWaaS firewalls, but also a common practice of most popular cloud firewalls configured in SaaS platforms (e.g., web application firewalls (WAF)). Furthermore, full traditional network security systems gradually become less popular on the security side because this is generally more expensive in terms of devices and design than for cloud systems (Figure 4)

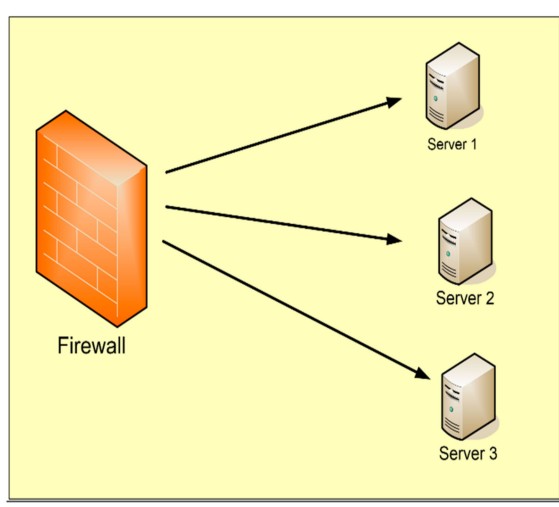

(**a**) Traditional Firewall Systems

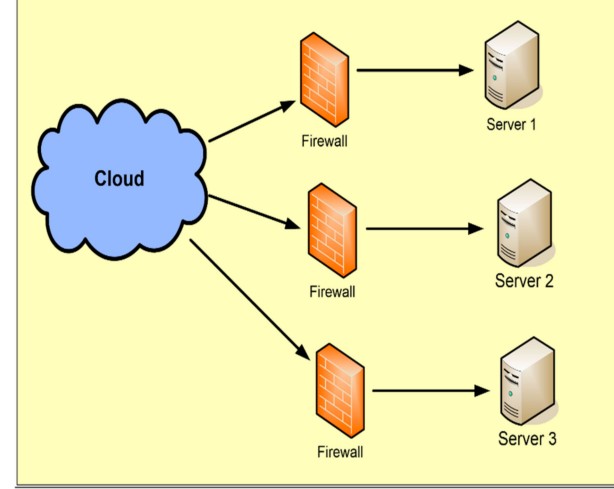

(**b**) Typical Cloud Firewall Systems (basics)

**Figure 4.** Cloud vs. Traditional Firewall.

FWaaS also introduces the innovative capability of extending the perimeter to all recognized users, regardless of the location they are based on (generally in tandem with Vir-

tual Private Networks), preserving the business for distant branch locations. Furthermore, following the principle of the cloud, FWaaS applies for pay-as-you-go plans, as a relevant cost-effective solution. However, despite all these advantages, the FWaaS is mainly deployed in IaaS and PaaS models, while SaaS does not generally deal with this firewall type. This introduces the first difference between those cloud firewalls mainly dedicated to IaaS and PaaS platforms, such as next-generation firewalls (NGFW), and others mostly used in SaaS platforms, such as a WAF. While FWaaS cannot be purely considered as a firewall type (because it is, rather, considered as a sub-category of NGFW), WAF and NGFW are the major modern-fashion firewall categories currently used in the cloud [29]. However, since virtual firewalls (VF) are in some way considered precursors to these technologies, then we start with this category and we will include them in our list.

### 3.2. Virtual Firewalls (VFs)

The growing use of virtual environments applied to the enterprise network architecture paradigm have widely led to the demand for virtual firewall devices (Figure 5), a software acting as a hypervisor in VMs or kernel modules, instead of physical devices [30,31], which is essentially addressed to those network systems more oriented to virtualized environments. Virtual firewalls were the first generation of devices no longer applicable at the hardware level, but directly within a virtualized environment, typically virtual machines (VMs), finally moving the same solution to cloud computing services. Next-generation firewalls (NGWF) are typically implemented in either PaaS or IaaS systems, while web application firewalls (WAF) are generally found in SaaS platforms and they are sometimes integrated into SaaS firewalls [32,33]. These firewalls are described briefly in the following subsections.

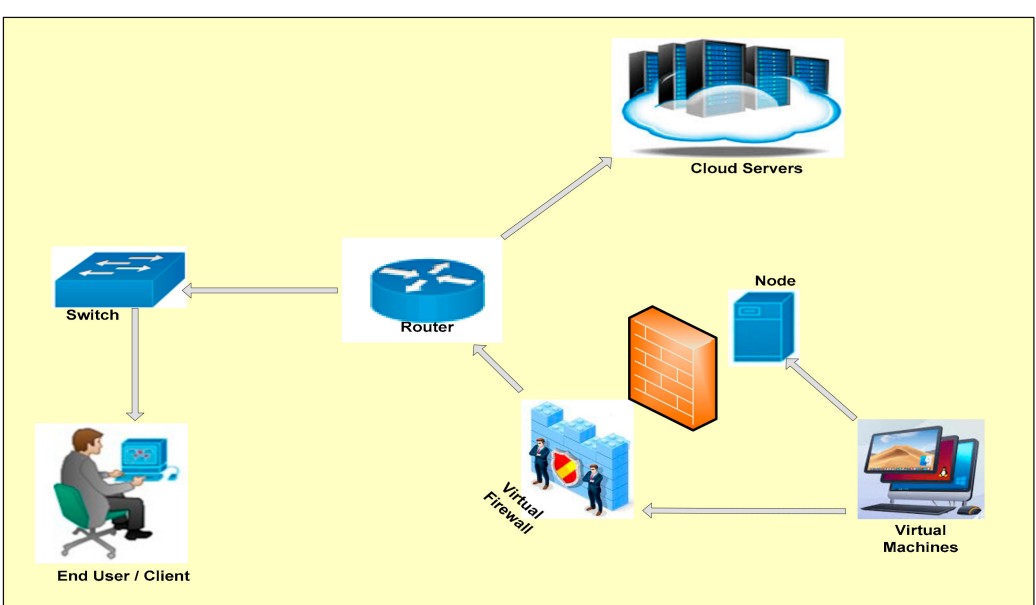

**Figure 5.** Virtual Firewall Network Topology.

Unlike the other category of cloud firewalls, virtual firewalls apply transversally to all cloud platforms. Virtual firewalls can also protect a broader range of machines, from a single server to a group of virtual servers (this practice is better known as micro-segmentation), applying all firewall rules and security policies automatically to all devices added to the firewall application. In contrast, there are some potential downsides. Firstly, a virtual firewall might not be easy to implement in a Wide Area Network using wireless devices or software-based technology, which requires constant software updates and maintenance support; secondly, like in traditional IT network systems, there is not enough room for customization. Thirdly, the high quality of design standards required may expose

cloud networks and security systems to additional (and sometimes unpredictable) design extra charges, and technically speaking, to potential vulnerabilities not easy to fix, especially when systems are hybrid cloud systems, a solution very often taken by those companies gradually moving to the cloud [34,35]. However, virtual firewalls are still smart and portable solutions specifically for smart healthcare, showing very useful benefits, such as:

- Better potential network performances in speed and connectivity terms due to more flexible positioning attitude and less resource consumption. A best practice can be to set a firewall machine behind each of the VMs, working as a managed kernel process for all running VMs separately [36]. This configuration should avoid the potential risk of malicious attacks to VMs when sharing network, storage, and computing resources.
- Virtual firewalls have additional functionalities provided by inter-VLAN (routed) firewalls and NIC-level firewalls, which are directly attached to VM NIC, able to filter ingoing and outgoing traffic through single VMs.

### 3.3. Next-Generation Firewalls (NGFW)

This category of firewalls became popular for the growing widespread use of sophisticated applications and the malicious threats involving them, but also for the lack of adaptability of traditional firewalls in recognizing different categories of web activity, in particular, the most recent ones requiring more flexibility when using permitting or rejecting rules within network traffic management. PaaS and IaaS platforms are common service delivery models to whom they are addressed, because of their high-security requirements [37]. On the one hand, if NGWF is a piece of equipment able to enforce network security policies through the application of all traditional firewall capabilities introduced before, in a sort of all-in-one body set of rules (standard mode) [38], the implementation of sophisticated intrusion prevention systems (IPS), the ability to interact as an autonomous body from OSI level 2 to level 7 of OSI layer (advanced mode), makes NGFW a powerful firewall, on the other hand. Other capabilities of this category of firewalls are website filtering, antivirus inspection, QoS, bandwidth analysis, and antivirus abilities. Moreover, NGWF is able to make whitelists and blacklists the function of their daily traffic reports on cloud services, finally adapting their rules to that specific need [38]. However, to widely protect and inspect the application layer with high levels of network traffic granularity, NGFW must add, in the future, some extra features currently not yet fully covered, such as encrypted traffic control, port hopping, application and identity-based control, URL filtering, data leakage protection, enforced WAN routing, and stronger QoS optimization capabilities, Wi-Fi network control, and accomplished network security policy for any connected device. Furthermore, NFGW has often been considered a very complex system, requiring high investments to be effective with some functionalities not necessary to all organizations [39].

### 3.4. Web Application Firewall (WAF)

The web application firewall (WAF) is a security solution at the application layer of the OSI model. This firewall was developed specifically for web security and plays a crucial role in securing the standard infrastructures, such as web servers and web applications, session handling and cookies, and other components typically part of a web application, also providing log monitoring tools and other security methods, such as URL encryption. Among all countermeasures, WAF has been able to take against most common attacks generally involving the web application layer, we can find logging secure checks, Secure Socket Layer (SSL) enforces encryption, data validation against buffer overflow, and many others, such as SQL, LDAP, XML, code injection cross-site scripting and tracing, HTTP request smuggling, and the aforementioned cookie protection [40,41]. In recent years, with the growing importance of cloud networks, the web application firewall has been extended to the cloud-based WAF, preserving characteristics, but with an elastic, scalable, and pay-as-you-go fashion. All of these favorable aspects are balanced by some risks requiring more complicated troubleshooting (e.g., when occurring false-positive events) [42]. Both

cause negative business impacts and cost opportunity disadvantages. Because of the importance of securing the web and the internet in cloud-based networks, in this section, we will discuss more in detail WAF firewalls and security practices, according to the environment where WAF is typically deployed. WAF essentially acts in synergy with other firewall software in the following five modes, according to the security policy applied to the architecture deployed [43]:

- Layer 2 bridge: in this mode, the WAF uses SSL decryption, and then is able to set a direct block of illegitimate packets through the WAF device. This allows global good performances, but despite these advantages, WAF cannot decrypt traffic with Diffie–Hellman techniques.
- Reverse Proxy: layer 2 bridge and reverse proxy mode are very similar to each other (Figure 6). However, in this mode, the WAF has its IP address and can decrypt, monitor, and check all web traffic. The WAF operates as a device between the firewall and the webserver.
- Embedded WAF: in this scenario, the WAF operates as an independent application (Figure 7a), working as software within the webserver perimeter.
- Out-of-band: unlike reverse proxy, in this mode, WAF is not running in the infrastructure.
- Internet hosted and cloud: it acts as a software, in the service delivery model, software as a service (SaaS) in the Cloud. WAF works in reverse proxy mode, with the DNS pointing directly to the cloud provider (Figure 7b). However, every policy should be checked and reviewed separately as the WAF is external to the corporate environment [44].

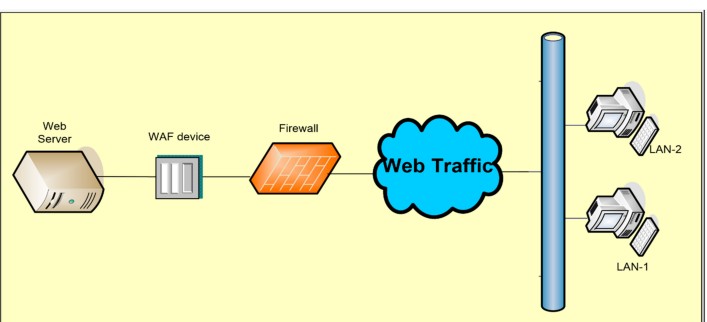

**Figure 6.** Reverse proxy and layer 2 bridge topology.

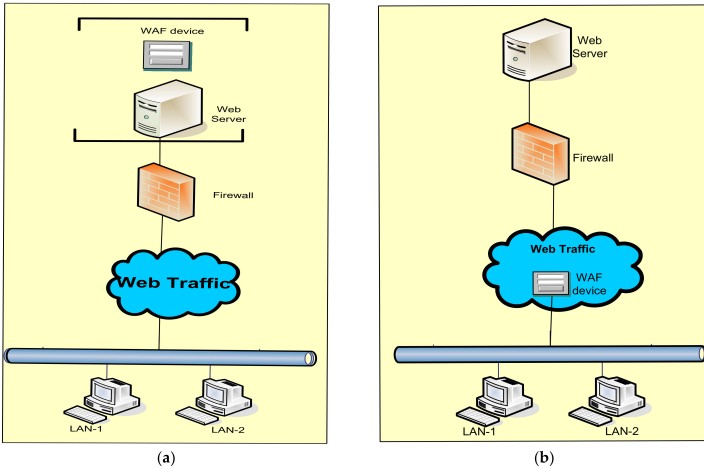

**Figure 7.** (**a**) Embedded web application firewall (WAF) topology, (**b**) internet hosted and cloud WAF topology.

This later classification makes it possible to distinguish these firewalls, which are intended for standard environments, from others that handle additional services capable to integrate cloud platforms. For example, next-generation firewalls offer advanced attack detection and removal features for cloud environments where there are generally more advanced categories and a vast set of attacks. Although this separation, some traditional firewall types, such as stateful inspection firewalls, may also operate in cloud environments since stateful inspection enablement is generally still preferred today and this separation is not necessarily intended for the targeted environments, but essentially due to topology constraints [45,46].

### 3.5. Traditional Firewall Types

Traditional firewalls control the traffic that enters to or exits from a network. The following sections describe traditional firewall types in detail.

### 3.6. Packet Filtering

The simplest type of firewall is packet filtering. Packet filtering operates at the network layer of the OSI model (Level 3). By use of the Internet Protocol (IP), a set of rules are created and applied on the network connection to accept or discard the traffic. Packet filtering firewalls allow or deny traffic in network IP protocols (TCP, UDP, or ICMP), in the source and destination of IP addresses, including ports and other settings of TCP headers. This firewall is quite straightforward to configure and can help to discard all unnecessary traffic, such as malicious packets transmitted through dangerous ports and the source of a potential attack. However, since protocol rules of packet filtering firewalls are essentially based on IP addresses, this does not ensure full protection of the network. Another problem is that IP packet filters are generally stateless, which means that the packets are not able to preserve and remember any previous traffic information [47]. This is undesirable for two reasons: (a) no packet memory means we cannot manage and secure our network effectively (it is as if we want to scout an area, and the map deletes checkpoints every time we finish the exploration of that area). (b) A stateless packet can be effortlessly spoofed due to the ACK bit in the packet's header and to the source address, then becoming potential targets for an attack [48]. For these reasons, this type of firewall is effective only when applying basic network rules directed to allow specific hosts (such as business partners) and to deny insecure hosts (i.e., blacklists).

### 3.7. Stateful Inspection

Stateful inspection firewalls operate at the session layer of the OSI model (Level 5). Unlike packet filtering, a stateful inspection can preserve all packet information previously processed and inspect information of ingoing and outgoing packets, managing threats from packet-to-packet simultaneously. For this reason, this firewall is often called a dynamic packet filtering firewall [49]. A potential benefit from the use of this firewall is the simple customization of the internal network, allowing network administrators to set specific ports (and traffic) they want to keep open. This prevents spoofing risks, and some hacking techniques, such as port scanning (but does not covering the application layer from SQL injections or buffer overflow risks). However, even though stateful inspection firewalls reach efficient levels of network security, some potential issues are the low level of network performance when not periodically supervised, maintained, and cleared [49,50]. This raises the important question between security and performance in every network architecture: the more we secure an environment, the more we must take care of network performances. Nonetheless, firewalls using stateful inspection rules are still applicable to a very large number of traditional and non-traditional security systems.

### 3.8. Application Proxy Firewalls

This type of firewall is the most efficient in current times. The application proxy provides security at the OSI application layer (level 7). It follows a stateful mechanism, but

unlike the stateful inspection, is also able to proxy the network acting as a gateway, with a man-in-the-middle behavior from server to client [51]. In other words, this firewall does not allow any packet directly sent from an application to the user and vice versa. Therefore, the application proxy can translate addresses and perform any additional control before allowing a connection to the server. The application proxy provides a deep packet inspection, preventing spoofing attacks and other hacking techniques, such as the stateful inspection firewall, but by keeping acceptable network performances. In addition to this, application proxy can block dangerous SQL commands and other web application malicious attacks, so providing an effective layer of security for different network topologies (such as cloud environments), optimal security isolation, and anonymity levels [26]. However, even though they have fewer maintenance requirements than stateful inspection firewalls, application proxies may entail bottleneck risks and other bad network performances if not periodically checked and updated. This category of firewalls typically applies to some specific categories of applications, such as e-commerce websites and other vendor platforms relying on web applications.

### 3.8.1. Circuit Level Gateway

This firewall affects the same layer of the stateful inspection firewall (session layer, level 5 of the OSI model). Potential advantages of using this firewall are the high level of protection and hidden information within private networks. This firewall can screen any (virtual) network extension, only allowing genuine connections. It is commonly used for VPN purposes and offers good levels of security protection in private networks [52]. However, the circuit level of the gateway does not filter single packets, so leaving network administrators to prefer stateful inspection firewalls rather than circuit-level gateway firewalls. Because of their proxy nature and following the same reasoning for the previous category of firewalls, circuit-level gateways can be used as stand-alone systems or within application proxies with specific types of applications.

### 3.8.2. Guard

The Guard firewall acts like a proxy firewall (OSI level 7), but in a more sophisticated manner, using advanced and complex rules. This firewall takes care of the quality of data, for example, by converting hot keywords to normal words. It operates using cryptographic rules adaptable to that specific environment, often making decryption hard to implement and generally compromising network performances for smooth changes of security rules [53]. Due to strict security constraints, which make guard firewalls too concerned with security rather than network performances, this firewall should be implemented in those systems, which are only focused on security.

### 3.8.3. Personal Firewall

The personal firewall is the last sub-category of proxy firewalls operating at OSI level 7, being an application recently used in SaaS cloud environments. It acts like a subnetwork defender, but only in a workstation, meaning that this type of firewall can manage single or multiple hosts, but only within the same network. Being often used as proprietary-based software integrated into cloud platforms, personal firewalls generally have limited network customization facilities [54].

Effective firewall practices can drastically reduce the risk of internal and external attacks and ensure a trade-off between security and performance within a network. Table 1 provides the pros and cons of traditional firewalls and cloud-based firewalls. However, benefits provided by firewalls at this stage are mainly structural advantages, meaning that, due to the respect of good practices when they have been created or at an architectural/topology level (theoretical protection), then they have the best ratio in terms of hard vulnerabilities.

**Table 1.** Summary of firewall types and their evaluation with pros and cons.

| Generation | Firewall | OSI Layer | Pros | Cons |
|---|---|---|---|---|
| Cloud Firewall | Virtual Firewall | Layer 2 | Applicable transversally on cloud platforms. Provides protection on a broader range of machines. | Implementation constraints in WAN networks using Wi-Fi devices/software-based technology Extra design charges required to avoid network exposure. |
| | Next-Gen. Firewall | Layer 2-7 | Wider protection at OSI layer, compatibility with Intrusion Detection System (IDS)/Intrusion Prevention System (IPS), Access Control List capabilities, advanced threat intelligence. | Too much complexity and high investments required. Some functionalities are not intended to standard network environments (cloud and not cloud). |
| | Web Application Firewall | Layer 2, 7 | Very specific solution against major malicious attacks at application level with adaptation capabilities for the target layer. Encryption and SSL force mode to establish secure connections. | Advanced troubleshooting skills required to avoid false positive patterns. |
| Traditional Firewall | Packet Filtering | Layer 3 | Easy to configure. Good packet processing efficiency. | Impossible to protect the entire network. Risk of attacks for misleading firewall configuration. Not able to filter at application layer. |
| | Stateful Inspection | Layer 5 | Able to manage multi packets and fewer open ports. Efficient management of threats and ability in blocking attacks, especially DDoS attacks, data packets memory. | High degree of skills to configure it Risk of network issue if not periodically maintained. Not effective in stateless protocols. |
| | Application proxy | Layer 7 | Able to "proxy" the network with a man-in-the-middle behavior. Deep packet inspection. Able to translate addresses, a comprehensive firewall system in tandem with Network Address Translation (NAT) and IDS/IPS. | Risk of network bottlenecks if not periodically maintained |
| | Circuit Level Gateway | Layer 5 | Screen all virtual network extensions in tandem with Virtual Private Network (VPN). Strong protection of private networks with good privacy rules. | It does not act as packet filter for individual packets. Requires network protocol changes (scarce adaptability). |

**Table 1.** *Cont.*

| Generation | Firewall | OSI Layer | Pros | Cons |
|---|---|---|---|---|
| Traditional Firewall | Guard | Layer 7 | Like proxy firewall with advanced and sophisticated rules. Cryptographic capabilities and can manage the quality of data. | Complex and with specific requirements. Much riskier in terms of network impacts. |
| | Personal Firewall | Layer 7 | Acts like a subnetwork defender. Compatible with SaaS delivery models. | It works only in a workstation with single or multiple hosts but within the same network. |

### 4. Firewall Placement in Smart Healthcare Environment

Firewall technology paved the way in the development of smart healthcare environments and provides smarter solutions in both network architectures and software domains [55]. The placement of the firewall is an important indicator for deciding on the best firewall type to use. Firewalls are generally used to protect network perimeters, typically at the WAN level. A firewall placed in the wrong layer can get a bad performance, even if the design of that firewall has been previously fulfilled [56]. The architectural adaptability, on the other hand, involves the minor or major difficulty of a firewall facing a network security architectural change. For example, if the network needs to add a new firewall or create a WAN connection, a suitable firewall is the one that can result in the least number of changes in the existing system. The firewall best practices have been classified using the following metrics:

- Placement;
- Architectural adaptability;
- Automatic update and reaction;
- Mode of operation;
- Privacy preservation.

Regarding the firewall placement, stateful inspection firewalls provide optimal performances while the connection has been established. WAF also offers the best performances when positioned closest to the targeted application and just behind load balancers when managing the security of multiple applications. Stateful inspection firewalls may also assume the form of application proxy firewalls under certain circumstances. The high level of architectural flexibility, scalability, and adaptability makes this firewall the preferred solution also for cloud environments [56]. Figure 8 depicts the high-level abstract of firewall best practices for a smart healthcare environment.

WAF is also particularly flexible in integrating existing architectures and can be configured in learning mode to make a step-by-step updated Access Control Lists. We have learned from the previous section that, in the application layer, we can find boundary checking errors as well as design and validation errors, then requiring additional network management capabilities [57]. In this regard, stateful firewalls keep any information about each connection state behind the firewall, ultimately informing the system if the test failed and provide potential ways to fix that issue and are successfully extended to heterogeneous networks characterized by modern security features and devices [58]. As a result, only already established sessions and previously filtered packets would be accepted by the firewall. Consequently, this modus operandi makes this category of firewalls the most preferred among all the other traditional firewalls. Concerning cloud firewalls, the fact that WAF acts directly to the web application and the application layer does not imply serious security constraint, therefore, WAF provides a certain degree of flexibility and attracts decision-makers to opt for this solution. Furthermore, WAF is particularly effective against several vulnerabilities at the data validation level, but they can also interact and

collaborate with the source code level, and then switch very quickly from denying rules to a recommended set of policy applicable in the next maintenance windows [59,60]. Some automatic rules (for example against data leakages) are configurable in last WAF application manager interfaces, such as the filtering of comments, which may involve sensitive area (i.e., passwords or other private content), and some parameters are automatically checked by the WAF, with regular ACLs update. Furthermore, WAF provides a self-test for quality assurance when new versions of the application have been released. Regarding privacy best practices, stateful inspection firewalls have some extra ability, and intelligent states allow manipulating information when applications are connected to it, such as the encryption mode. This is a robust security state allowing to preserve the information when new sessions have been established. Conversely, WAF firewalls can force SSL mode in the function of the encryption strength previously defined [61]. Following this survey, stateful inspection and WAF firewalls are generally the most advised category of firewalls for cloud environments. As seen in previous sections, they are neither exempt from challenges, further required improvements, nor extra features to broadly cover residual vulnerabilities, but concerning the application of best practices in these firewall categories, satisfactory results in targeted environments can be made.

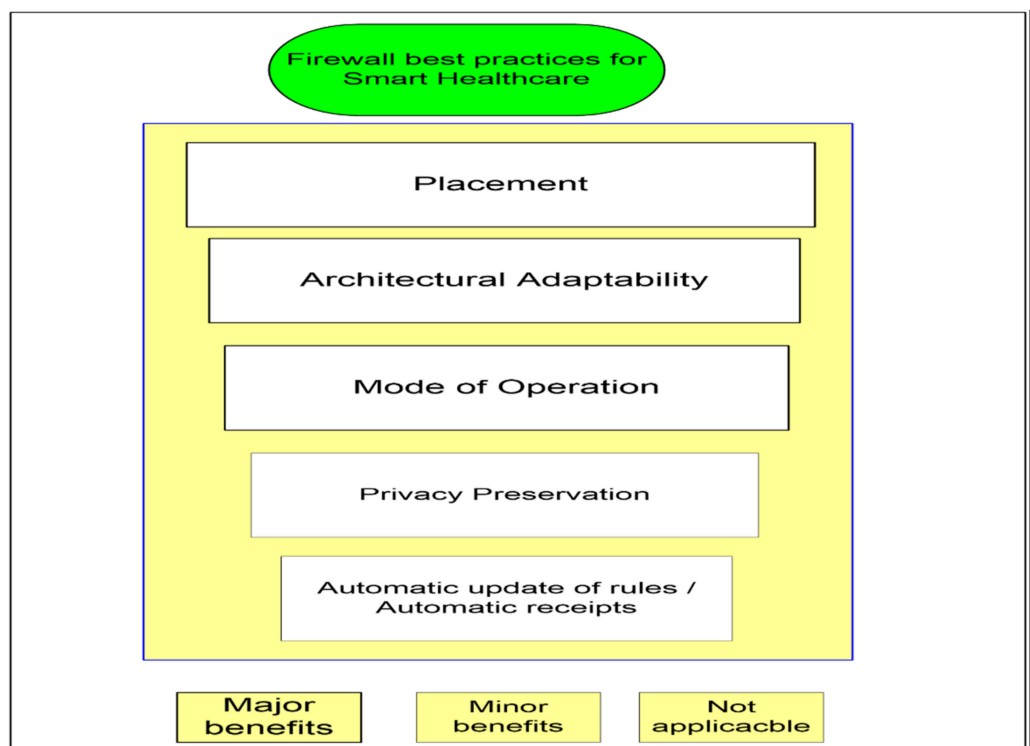

**Figure 8.** Firewall best practices for smart healthcare environment.

Moreover, it has been seen that cloud networks suffer a larger number of security threats than traditional networks, meaning that firewall policy and rules should be frequently reviewed and updated according to the requirement of smart healthcare environments. More specifically to the network security domain, autonomic computing invests an important role to promote best practices in security automation facing multiple threats and different defense models. Firewalls have also been impacted by this paradigm and best practices mainly involve the automatic update of firewall rules and policies. From a functional point of view, the operating mode of a firewall describes the two general different concepts for a firewall to act [62]:

- Not allowed means denied.
- Not specifically denied means allowed.

The first approach creates safe environments but gives fewer choices to the user for any future configuration change. In the second case, the firewall allows all traffic, and any potential issue is studied on a case-by-case basis. This approach creates a more flexible environment but requires more frequent updates, especially as the network grows in size and complexity. The best firewall can balance between these two operating modes. Finally, privacy preservation. With the advent of the General Data Protection Regulation (GDPR), respect for privacy has increasingly become a legal (as well as a security) aspect [63,64]. An effective firewall must be able to offer an advanced level of protection of personal data, such as protection in the use of search engines, protection against tracking scripts and tracking cookies, protection from digital surveillance, and other tools to protect your IP address. The security and prescriptions of the GDPR can also be implemented only if the system is properly configured to prevent unwanted access or exposure of public ports through the configuration of ACL, protect incoming connections by avoiding too permissive rules, and possibly filter outgoing connections by means of inspection filters applied on different layers. The constant updating and monitoring of equipment is also an important point.

## 5. Firewall Vulnerabilities

The classification of firewall vulnerabilities may help to find (quicker) any potential issue exposing a network system to security concerns throughout the use of that firewall policy. Common issues generally include wrong configurations, design errors, or misleading firewall configurations and settings; thus, disclosure of poor and unsatisfactory security policies [65]. Even worse, a misconfigured firewall may leave the system, which is supposed to be protected, exploited from the outside. Vulnerabilities, such as validation or design errors, are generally associated with IP and port filtering practices. When an error occurs, the probability of a malicious event or an attack (e.g., DDoS issue) increases accordingly. Moreover, a vulnerability belonging to the first layers of the OSI model (e.g., Layer 3) is still present in other layers touched by that application. In other words, layers are correlated with each other, and vulnerabilities cumulate throughout the layers. Vulnerabilities follow the fault classification scheme [66,67]. Spurious, misplaced, missing, and incorrect entities are erroring whose corrections may require an insertion, a change, or removal of an entity from the original source code to fix that vulnerability. Some common vulnerabilities affecting both hardware and software firewalls are troubles in rejecting policy for illegitimate network traffic, ignoring rules for malicious threats, lack of security rules for domestic network traffic, then causing insider attacks. Vulnerabilities typically affecting only software firewalls are any malicious action aiming to bypass the firewall by exploiting it at the application layer (application control, masquerade, prevent loading, firewall uninstall, etc.), or any attempt to move the application of the firewall at a lower level, such as the network layer [68].

Firewall vulnerabilities can be resumed into two big categories: (a) vulnerabilities related to design constraints and limitations; (b) vulnerabilities related to misconfigured firewall policy and rules. While in the second category, best practices can potentially fix the vulnerability assumed in that system (soft vulnerability), vulnerabilities of the first category are typically due to design/topology issues, then best practices can only be applied at the source (hard vulnerability), by providing, for example, a new design of the firewall [68]. Firewall engineers should always avoid the first category of vulnerabilities that happen after the testing stage of the product cycle has been validated. As a result, software firewalls disclosing hard vulnerabilities in production environments will no longer be used, and then uninstalled, and firewalls that are more effective will be considered. Table 2 provides the most frequent and common firewall vulnerabilities in the function of the firewall type(s) by cause, effect, and fix remedy of the issue found. For example, an issue, such as the validation error, occurring when a program is running in an environment regardless of the correctness of the data introduced in that environment (cause), is strictly related to a

DDoS issue (effect). Likewise, firewall design errors (cause) lead to issues in the execution of codes (effect) [69].

**Table 2.** List of most known firewall vulnerabilities and targeted firewall type.

| Firewall Vulnerability | Firewall Types Commonly Impacted |
| --- | --- |
| Authorization error | Application firewalls |
| Code execution | Application firewalls |
| Spurious entity | Any |
| Incorrect entity | Any |
| Validation error | Packet filtering |
| Domain error | Packet filtering |
| Target resource access | Packet filtering, Guard, Personal firewall |
| Weak or not correct design error | Any |
| Serialization error | Packet filtering |
| Alias error | Packet filtering |
| Other major logic error | Any |
| Target resource change | Packet filtering, Guard, Personal firewall |
| Misplaced entity | Any |
| Missing entity | Any |
| Boundary value error | Potentially all firewall categories |
| Denial-of-Service (DDoS) | Application firewalls, Packet filtering but potentially any firewall |

The role of firewalls in smart healthcare as a technical countermeasure has been progressively recognized in the security life cycle as an important tool to achieve security goals [70]. This work is a capstone for security and provides a clear categorization of firewall types and practices, also treating security for cloud environments.

Regardless of the firewall type and the technology used, a fully effective firewall would always protect against internal and external security weaknesses and provide the right trade-off between security and performance [42]. Conversely, the correct use of firewalls within a system can be a powerful and effective resource in terms of security protection.

## 6. Firewall Best Practices for Smart Healthcare Environment

The interconnectivity of a smart healthcare environment provides a seamless connection across multiple entities where a patient can get engage with doctors and other paramedics to better manage their care. Effective security measures can protect critical data and patient's privacy. Continual change in the healthcare environment influences the use of security equipment, especially the use of firewalls. Moreover, reliable, intelligent, and secure monitoring and management system reduces the impact of threat and risk to the smart healthcare environment. Firewalls not only protect smart healthcare environments from malware and viruses, but also defend against phishing and other cyber-security threats. Network firewalls filter the traffic between the secure network and the internet. Similarly, host-based firewalls run on the server, which control the network traffic. Moreover, host-based firewalls are suitable if patients are connecting through a remote network. Cloud-based firewalls are an important addition to the existing firewalls and provide more benefits in terms of security, more specifically, in securing the cloud-servers [71].

Smart healthcare environments require special security considerations due to the patient's data and other information. In addition, the various interconnected medical devices are often overlooked where hackers and adversaries can gain unauthorized access.

Firewalls are one of the most basic building blocks of securing the smart healthcare environment. Solid and properly configured firewalls repel the attacks and act as a foundation for a smart healthcare environment [72].

## 7. Discussion and Recommendations

In this paper, we provided a comprehensive classification of firewall types, best practices, and an update of firewall taxonomy for the cloud and of the most significant firewall vulnerabilities. Due to their ineffectiveness in securing networks and to high exposure to external attacks, packet-filtering firewalls are not generally recommended, and are not used in cloud firewalls. Stateful inspection firewalls are often implemented in modern security systems. It was also observed that a great variety of firewalls applied to traditional network systems are still valid for cloud environments, providing different levels of effectiveness, efficiency, and security, according to the way they operate within network systems. Vice versa, a new generation of firewalls, such as virtual firewalls, are essentially applied to virtual environments; they are still applicable to traditional networks. The virtual firewalls also provide a useful workaround for issues typically involving physical firewall devices, such as firewall rule anomalies. However, making perfect designs for virtual firewalls is still a challenging task for cloud environments due to the risks; cloud systems are exposed to design errors and weaknesses that are difficult to fix during the execution. Therefore, an interesting aspect involving cloud-based firewalls is to go beyond the perimeter and extend the role of the firewall as we usually think. Moreover, modern firewalls are stateful, providing an effective packet inspection, intelligently dropping all suspicious traffic. However, even the most powerful firewall needs to work with other networks, security components, and devices (for example Intrusion Detection System and Intrusion Prevention System). For instance, stateful inspection, proxy application, WAF firewalls, and other recent firewall types, can work in tandem with the most advanced hardware and software security devices, and are considered as best firewall practices.

In recent times, smart healthcare monitoring services are increasingly leveraging due to offering benefits specifically under current circumstances and pandemics the world is facing. Moreover, smart healthcare environments play a crucial role in the timely delivery of healthcare services and advice to patients and to professionals. Moreover, data captured through embedded sensors and wearable devices can reveal critical conditions about the patients and communicate through the internet and cloud servers. Adversaries target the healthcare environments to gain financial benefits through obtaining the patient's data. Integration of the cloud with the healthcare environment requires special security provisions to protect not only the privacy, but understand the risks as well. Similarly, additional security measures and control need to be in place along with the firewall placement. Moreover, delivering end-to-end security requires preserving patient data and privacy through the implementation of proper firewall technology. Therefore, it is needed to ensure the proper security, privacy, and availability of smart healthcare systems.

The emergence of healthcare-related cybercrimes is a major concern to healthcare systems where data and security breaches cause not only financial issues, but negatively damage reputations and impact litigation [73]. Another important area of concern is the rise of the internet of medical things (IoMT), where a large number of devices are communicating with other devices to form a network and to share personal data; hence, protection of data and securing the devices is another dilemma. Another growing area of threat is the attack on wearable and implanted medical devices, where a single point of failure in a centralized network will adversely impact a patient's life.

In our previous [74] work, we identified the challenges for implementing a combined healthcare based platform and proposed a blockchain based solution to address some of the challenges. In the future, we plan to extend our previous work and implement the firewall security recommendation for securing the implementation of the healthcare system.

## 8. Conclusions

This review paper covers the various security threats and challenges facing smart healthcare environments and provides a comprehensive classification of firewall types and valuable best practices, including implementation strategies and vulnerabilities. The smart healthcare sector is growing increasingly, and the use of online consultations from paramedics and preserving a patient's privacy and data is a real challenge, where a large amount of personal data is to be managed. Moreover, the provision of remote patient monitoring and collecting data pose a series of challenges to the smart healthcare environment. However, this review paper identifies a wide range of firewall types and highlights the benefits and drawbacks. Therefore, the selection of firewalls that are suitable for smart healthcare environments, and to get the maximum benefit, is equally important with proper implementation of plans and policies. Moreover, we are planning to address the role of intrusion detection and intrusion prevention systems (IDS/IPS), including cloud-based systems for smart healthcare environments, in our future work.

**Author Contributions:** All of the authors equally contributed to preparing the manuscript. T.A. devised the methodology and supervised the work. R.W.A. and F.P. drafted and edited the manuscript. All authors have read and approved the final version of the manuscript.

**Funding:** The authors would like to thank Arab Open University—Muscat, Sultanate of OMAN, for supporting this work.

**Acknowledgments:** The authors are extremely thankful to Arab Open University (AOU), for its esteemed support and encouragement.

**Conflicts of Interest:** The authors declare no conflict of interest.

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
