# Peer review of "Firewall Best Practices for Securing Smart Healthcare Environment: A Review"

_applsci, doi:10.3390/app11199183_

Round 1

Reviewer 1 Report

The authors have produced a fairly comprehensive overview paper that addresses the various security threats and challenges.

Their attention is focused on intelligent service environments in healthcare.

The paper provides a comprehensive classification of firewall types and valuable best practices including implementation strategies and vulnerabilities.

At times, the authors find it difficult to maintain the style of a review paper, and may stray into the style of a textbook.

It would be nice if more tabular comparisons could be made and the studies mentioned could be discussed for and against.

Further remarks

Figure 1 is not consistent and logical. Lines run into the doctrine. The components are not logically connected and the components are not consequently labeled.

The caption is not meaningful. How do the authors come to the conclusion that this is a Typical Smart Healthcare Architecture?

Fiure 2 How do the authors come to the classification of near traditional and cloud based firewalls?

Figure 7 Adjusting the alignment of the pictograms to the text

Author Response

Kindly refer to the attached file.

Reviewer 2 Report

The authors give a very detailed overview of firewalls in general and firewall types, their advantages and disadvantages. Also, they touch the subject of firewall usage in smart healthcare environments. However, most of this paper focuses on firewalls in general and does not bring any scientific novelty to firewalls nor to the (smart) healthcare environments. If the paper really wants to focus on usage in healthcare, the authors should shift the focus to specific problems in this domain and try to propose a novelty with specific use of firewalls in this domain. Unfortunately, this does not seem to be the case in this paper, since it just gives an overview of firewalls in general. A very good overview, but I do not see any scientific relevance in this paper as such.

Some other remarks:

- Why do you focus only on firewalls? Firewalls are merely one and probably the most simple line of defence, the environments you are mentioning could perhaps gain bigger benefits with more advanced IDS/IPS network or cloud based systems.

- How does smart healthcare architecture differ from other similar architectures?

-  The paper contains many spelling errors, please revise the complete paper!

- “Malwares” vs Viruses, Worms and Trojan attacks - these are all malware

- Page numbering on tables is wrong?

- You should focus more on problem definition in healthcare environments and then describe how exactly firewalls could be used for solving some of the presented problems

Author Response

Kindly refer to the attached file.

Reviewer 3 Report

This paper attempts to summarize various firewall types with benefits and drawbacks from healthcare perspective.

Main purpose of this paper is to define comprehensive set of policies for securing smart healthcare devices.

There are several issues to be addressed for publication.

  1. First of all, this paper lists up various types of firewalls from many different points of view.  I suggest that authors include more in-depth analysis of each firewalls with qualitative and quantitative evidences or examples rather than presenting anecdotal arguments.
  2. Also, it would be helpful to provide advice and guidelines for implementing secure smart health devices based the review the authors made in this paper.
  3. Specifically, it would be better to further extend section 7, which is about research directions for securing healthcare devices.

Author Response

Kindly refer to the attached file.
